# [Regular] RuleSum: Injecting Rulesets into Knowledge Graphs for Accurate and Accessible Legal Summarization

**Saumya Chauhan**[*]
California Institute of Technology
schauhan@caltech.edu

**Aditi Chandrashekar**[*]
New York University
ajc10180@nyu.edu

## Abstract

Legal texts are often complex and inaccessible, limiting understanding for non-experts. Large language models (LLMs) can summarize such material but often sacrifice interpretability and factual accuracy. Thus, we present **RuleSum**, a framework that integrates structured rulesets and knowledge graphs (KGs) with LLMs to generate legal summaries that are faithful, readable, and pedagogically aligned. Leveraging the IRAC method (Issue, Rule, Application, Conclusion) as a reasoning scaffold, RuleSum applies structured representations—free-form, tuple-style (KAPING), and IRAC-labeled serialization—to guide summarization. We provide a framework for evaluation along multiple axes, including semantic consistency and readability. We evaluate our approach on the MultiLexSum dataset, using ROUGE-L for lexical overlap with reference summaries, SBERT for semantic similarity, and Flesch–Kincaid Grade Level (FKGL) for readability. The IRAC-guided KAPING-IRAC configuration consistently outperforms all baselines, achieving the highest alignment with reference summaries while maintaining accessibility for general audiences. Finally, we provide an interactive Gradio-based demo and open-source code (github.com/Saumya-Chauhan-MHC/Rulesum), that visualizes how each pipeline stage improves clarity and factual grounding, supporting future applications of structured reasoning for education and decision-making.

## 1 Introduction

Legal documents are notoriously complex, making it difficult for students and non-experts to identify and understand critical information. While large language models (LLMs) like GPT-4 can produce fluent summaries of such documents, their reasoning processes remain opaque and rarely follow the formal structure expected in legal analysis [1]. In high-stakes domains such as law, where accuracy and clarity are paramount, this lack of interpretability (e.g. opaque reasoning chains) and potential for factual error (hallucinated or imprecise details) greatly limits the usefulness of raw LLM-generated summaries. Learners cannot easily discern how an LLM arrived at a conclusion or verify the facts in its summary. These limitations highlight the need for summarization frameworks that provide not only concise overviews, but also transparent reasoning and faithfully grounded content. Existing LLM-based approaches fall short in educational settings, motivating new methods that align summaries with formal legal reasoning to improve interpretability and trust.

One approach to improve factual alignment and transparency is to augment LLMs with structured knowledge. Prior work has integrated knowledge graphs (KGs) and extracted fact triples into the summarization or QA process to better ground outputs in verifiable information. For example, the KAPING system retrieves and verbalizes relevant KG triples to assist in zero-shot question

---

[*]Equal contribution.

answering, and KG-to-text frameworks rewrite subgraphs into fluent, answer-oriented sentences – underscoring how the serialization format of facts can influence a summary's factual accuracy [2–4]. Beyond retrieval-based augmentation, systems like RTSUM rank triples by salience to improve interpretability, and graph condensation methods (e.g., PCSG, HCSumm, KGTrimmer) preserve key semantics in a compact form [5–8]. However, even these structured augmentation methods do not explicitly incorporate domain-specific reasoning models or educational scaffolds. In the context of legal education, a summary needs to explain why and how conclusions are reached, not just state the conclusions, which calls for integrating formal reasoning patterns into the summarization process.

The legal domain introduces additional challenges and opportunities for structured summarization. Legal reasoning often follows a formal schema taught in law schools, such as the IRAC framework (Issue, Rule, Application, Conclusion), which provides a canonical scaffold for argumentation [9]. Datasets like MultiLexSum align lengthy case documents with expert-written summaries (including issue and holding annotations), underscoring the need to capture not just facts but the logical flow of arguments in a case [10]. Incorporating an IRAC-based scaffold into the summarization pipeline can therefore help align generated summaries with the way legal reasoning is communicated by experts. Yet, few existing summarization systems fully leverage such formal reasoning schemas, limiting their effectiveness as educational tools for illustrating the step-by-step logic of legal decisions. This gap suggests that a combination of structured knowledge and domain-specific reasoning cues could significantly improve the interpretability of legal summaries.

In this paper, we propose RuleSum, a structured summarization framework that injects rulesets and knowledge graphs into the LLM's generation process to produce legal summaries that are faithful, readable, and pedagogically aligned. RuleSum leverages the IRAC method as a reasoning scaffold and applies multiple representation strategies – from free-form text rephrasings to tuple-style fact triples (KAPING format) and IRAC-labeled text segments – to guide the LLM in breaking down complex legal language [2]. By unifying symbolic knowledge with the generative fluency of LLMs, this design enables the model to capture not only the content of a case, but also the logical progression of its arguments in the summary. We evaluate our approach on the MultiLexSum dataset, using metrics that assess both factual fidelity and accessibility (ROUGE-L for overlap with reference summaries, SBERT for semantic similarity, and Flesch–Kincaid Grade Level for readability) [11–13, 10]. In experiments, an IRAC-guided configuration of RuleSum consistently outperforms a baseline GPT-4 summarizer, achieving higher alignment with expert reference summaries while maintaining a lower reading level (i.e. improved readability for general audiences). These results demonstrate that injecting structured knowledge and legal reasoning cues into the summarization process can substantially improve both the accuracy and clarity of the generated summaries.

Our contributions are the following:

1. **A novel legal summarization framework that decomposes complex legal language using multiple structured representations and an IRAC-guided pipeline.** RuleSum is the first framework (to our knowledge) that explicitly integrates the IRAC reasoning schema into both the knowledge graph construction and the LLM prompting stages of summarization, aligning the summary's structure with formal legal reasoning steps. The framework uniquely combines several representation methods – including free-form text extraction, knowledge-graph triples, and IRAC-labeled text segments – at different stages of the pipeline to break down dense "legalese" into more accessible language without losing the case's logical structure. By using IRAC-based quotas in triple selection and organizing facts under Issue/Rule/Application/Conclusion labels, our approach ensures that each part of the summary corresponds to a reasoning role, which sets it apart from prior legal summarization systems that lack such structured scaffolding [2, 5].

2. **A comprehensive evaluation methodology that disentangles readability and factuality in zero-shot legal summarization.** We design a factorial study that (i) controls summary length to probe completeness–clarity trade-offs, (ii) varies triple-selection policies to contrast reasoning-role coverage with semantic relevance, and (iii) ablates structured components across the pipeline—KG use, serialization, and a no-knowledge baseline. This yields 48 configurations per case and ∼1,200 summaries overall, evaluated with ROUGE-L (lexical alignment), SBERT (semantic similarity), and FKGL (readability), plus readability-quartile analyses to assess behavior on harder texts. Beyond reporting aggregate gains, this protocol isolates each module's contribution and surfaces regime-specific strengths (e.g., IRAC-quota + KAPING-IRAC under higher FKGL), providing a reusable template for rigorous,

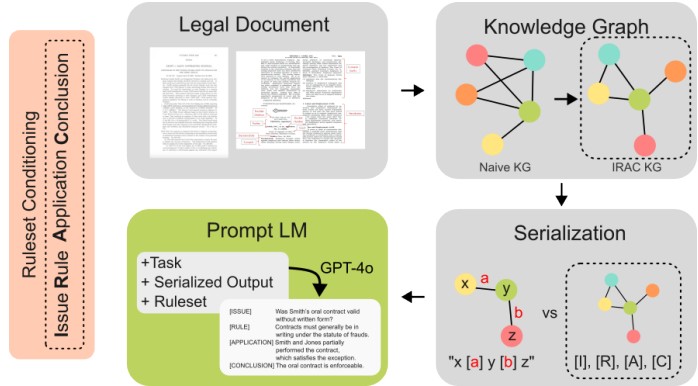

Figure 1: RuleSum pipeline: a legal document is parsed into a knowledge graph (KG) of fact triples, serialized, and injected into a GPT-4o prompt OpenAI [1]. The IRAC framework can guide graph construction or prompting, followed by optional refinement.

component-wise evaluation of structured LLM summarizers in education-oriented, high-stakes domains [11–13].

3. **An interactive visualization tool and demo that enhances transparency in the summarization process.** We developed a visualization interface allowing users to explore which parts of the source text and which knowledge graph triples contribute to each section of the generated summary. This interactive demo shows how a particular Issue sentence in the summary is grounded in specific source passages or facts in the graph. Such transparency allows users (e.g. law students) to trace the origin of each summarized point, building trust in the summary's accuracy. This insight helps users learn how to better formulate prompts and questions for LLMs in similarly complex domains as they can see the effects of structured guidance on the output. We have open-sourced the code and demo for RuleSum, enabling others to examine the step-by-step reasoning integration and supporting future applications of structured reasoning in education and decision-support [9].

## 2 Background and Related Work

**LLM Summarization with Structured Augmentation.** To improve factual grounding in specialized domains, recent work augments LLMs with knowledge graphs (KGs). KAPING retrieves and verbalizes relevant triples for zero-shot prompting, boosting factuality without fine-tuning [2]. RRA instead retrieves a subgraph and rewrites it into coherent text for the prompt, showing that *how* KG evidence is serialized (triples vs. narrative) strongly affects downstream quality [4].

**Triple Selection, Salience, and Interpretability.** Selecting which facts to include is crucial at scale. RTSUM ranks relation triples using multi-level salience and then "sentencifies" top items, yielding concise outputs with traceable evidence links—an interpretability benefit directly relevant to legal settings [14].

**Graph Condensation and Pruning.** KG summarization/pruning methods target compact yet representative subgraphs. PCSG optimizes pattern coverage and connectivity for RDF snippets [15]; HCSumm preserves latent structure by approximately maintaining embedding distances [16]; and KGTrimmer prunes nodes/edges by dual-view importance for recommendation without hurting accuracy [8]. Our use of top-$k$ triples differs: we condense *for reasoning coverage*, not maximum compression or structural preservation [15, 16].

**Serialization Formats for LLMs.** KG evidence can be injected as compact tuples (KAPING-style) [2] or rewritten sentences (RRA) [4]; the former is precise, the latter easier for LLMs to assimilate. We compare these with an IRAC-labeled format that groups facts by reasoning role, explicitly scaffolding the argumentative flow.

**Positioning of Our Approach.** RuleSum combines KG augmentation with a domain-specific schema: IRAC labels guide *both* selection (IRAC-Quota) and serialization, complementing relevance/salience-

127 based selection (e.g., RTSUM) and prior KG-to-text prompting (e.g., KAPING/RRA) [2, 4, 14]. This
128 alignment to legal reasoning is orthogonal to general graph condensation goals [15, 16].

# 3 Proposed Solution

## 3.1 Problem Formulation and Workflow

Legal case texts are not only long but also follow a formal reasoning structure, making them challeng-
ing for non-experts. Large language models (LLMs) like GPT-4 can produce fluent summaries, but
their reasoning is often opaque and unstructured. To bridge this gap, RuleSum introduces a pipeline
that integrates knowledge graphs (KGs) and formal rulesets into the summarization process for better
factual accuracy and interpretability. We leverage the IRAC method (Issue, Rule, Application, Con-
clusion) as a reasoning scaffold – a canonical structure in legal analysis – to guide the summarization
towards legal pedagogical norms.

**Workflow:** As shown in Figure 1, our system first parses the input legal document into a KG of
factual triples (subject, relation, object) capturing key entities, events, and outcomes. This KG
(optionally enriched with IRAC labels) is then serialized into text and injected into the prompt of an
LLM (GPT-4). Finally, the LLM generates a summary conditioned on this structured context. The
design unifies symbolic structure with LLM fluency, allowing the model to capture not just relevant
content but also the logical progression of arguments. In essence, RuleSum's pipeline ensures that the
summary remains faithful to the case facts and accessible in explanation, by combining the strengths
of knowledge encoding and natural language reasoning. The following subsections describe each
component and its motivation in detail.

## 3.2 Knowledge Graph Construction

The first stage is to construct a knowledge graph from the legal text. We employ an LLM-assisted
extraction module (built with LangChain's graph-extraction tools [3]) to identify salient entities,
relations, and facts from the case document. Each fact is represented as a triple (e.g., Party_A – wins
– claim_X), forming a case-specific KG. This approach builds on prior work in knowledge-augmented
prompting, such as the KAPING system which retrieves and verbalizes triples to assist QA models.
By distilling the source text into a set of triples, we create a compact, structured representation of the
case's key points.

Crucially, RuleSum can incorporate IRAC-guided knowledge graphs to inject domain reasoning. If
IRAC integration is enabled, the extraction step tags each triple with one of the IRAC roles – Issue,
Rule, Application, or Conclusion. For example, a triple describing the central legal question would
be labeled as an Issue, whereas a triple stating the court's decision would be labeled as a Conclusion.
This IRAC-tagged graph provides a scaffolded view of the case: it not only lists facts, but also situates
them in the argument's logical structure. If IRAC labeling is turned off, the KG remains unstructured
(a plain set of triples). By encoding legal reasoning roles into the KG, we enable the summarizer to
more readily follow the flow of a legal analysis, rather than treating all facts as equal or disconnected.
This step is inspired by the formalism of rhetorical role annotation in legal texts and ensures that our
framework aligns with how experts write case analyses.

## 3.3 Triple Selection and Serialization

Given the full KG (which may contain dozens of triples), the next challenge is to select and linearize
the most relevant facts for the LLM. Including every extracted triple could overwhelm the prompt
or introduce irrelevant details. We therefore experiment with two complementary top-$k$ selection
policies for choosing a subset of triples:

**IRAC-Quota selection:** This policy ensures balanced coverage of each IRAC reasoning role. We
allocate slots for triples such that Issue, Rule, Application, and Conclusion facts are all represented
in the prompt. For instance, if $k = 8$ triples are to be used, the quota policy might select the top 2
triples from each IRAC category (assuming enough triples exist per category). This guarantees that
the summary isn't missing an essential part of the legal reasoning, addressing interpretability and
completeness. Our IRAC-Quota strategy extends prior work on guided summarization – whereas

RTSUM ranked triples by general salience, our method explicitly balances salience with logical role coverage to preserve the case's argumentative structure.

**Similarity-based selection:** This policy ranks candidate triples by their semantic relevance to the source document (or a reference summary) using embedding-based similarity. The intuition is to pick facts that best capture the core content of the case. This approach connects to established graph condensation methods like PCSG and KGTrimmer, which aim to retain the most informative parts of a knowledge graph. By using sentence embeddings, we prioritize triples that cover important concepts and events from the case, serving a role similar to how prior systems select supporting facts for QA. We include this policy to benchmark a purely relevance-driven selection against the more structured IRAC-Quota strategy.

After selection, the chosen triples must be serialized into a textual form that can be fed into the LLM prompt. We implement three serialization formats in the RuleSum pipeline:

**Free-form verbalization:** Each triple is converted into a natural language sentence, yielding a short paragraph of text. For example, a triple (Defendant – owed – duty) might be verbalized as "The defendant owed a duty." This free-form mode aims for readability and seamless integration into the prompt, at the cost of possibly losing the explicit triple structure. It aligns with approaches where KG facts are rewritten into narrative statements for the model (e.g., the RRA Retrieve-Rewrite-Answer framework that turns subgraphs into answer-oriented sentences).

**Structured tuple (KAPING) format:** Triples are presented in a terse, tuple-style notation. We adopt the format used by KAPING, which might list facts as (Subject; Relation; Object) or a similar semi-structured template. For instance: (Defendant; owes; duty of care). This compact representation preserves the precise subject-predicate-object structure, providing the LLM with clear symbolic facts. Prior work has shown that such structured prompts can improve factual grounding in zero-shot settings. However, tuples alone lack explicit cues about their role in the argument.

**IRAC-labeled serialization (KAPING-IRAC):** This mode groups the selected triples under IRAC section headers. By prefixing each group of facts with its IRAC category, we provide the model with strong hints of the legal reasoning context for each fact. This format, essentially KAPING tuples organized by IRAC, offers explicit reasoning cues while retaining symbolic precision. We expect this to guide the LLM to produce summaries that follow the Issue-Rule-Application-Conclusion narrative, improving interpretability. Notably, our use of IRAC labels in serialization is a novel addition building on the idea that structural organization of input can steer an LLM's output style. It goes beyond surface-level fact inclusion by embedding an outline of the argument within the prompt.

By comparing these three formats, we explore the trade-offs between a more natural language context (free-form), a concise factual context (tuple format), and a structured reasoning context (IRAC-labeled). Past studies underscore that how knowledge is presented in the prompt greatly affects factual grounding, so this component is key to RuleSum's design.

## 3.4 Prompt Engineering and Summary Control

The final component of RuleSum is the prompt engineering that integrates the selected and serialized facts into the LLM's input and controls the summary generation. We use GPT-4 as our base summarization model. The prompt to GPT-4 includes two main elements: (a) the knowledge-graph content (in one of the serializations above) and (b) instructions specifying the desired summary style and length.

**Length control:** We direct GPT-4 to produce summaries in one of three target length ranges to examine the effect of compression. In particular, we define a tiny summary (50 words), a short summary ( 150 words), and a long summary (300 words). The prompt explicitly mentions the target length (e.g., "Summarize the case in about 150 words..."). By varying length, we can study the trade-off between informativeness and readability: shorter summaries force the model to be concise, while longer ones allow more details at the risk of increased complexity. We found that GPT-4 adheres to these length instructions reliably, which is crucial for fair evaluation. Each length setting serves a different educational use case – from a quick gist (tiny) to a detailed study aid (long) – and testing across them provides insight into how our structured approach scales with summary depth.

**Prompt structure and summary control:** In the prompt, the chosen facts (free-form sentences or tuples with optional IRAC headings) are typically prefaced by a brief instruction (e.g., "Facts: . . .")

and followed by the summarization cue (e.g., "Summary: Please provide a summary of the case..."). This ensures the model treats the injected KG facts as contextual information and not as the final answer. By labeling the sections (and using IRAC headings in one mode), we nudge the model to organize its output in a logical order. The combination of factual grounding and IRAC cues in the prompt is designed to produce summaries that are both faithful to the case facts and aligned with the pedagogical structure of legal reasoning. Any hallucinations or deviations from the facts can be minimized since the model can rely on the injected triples as a source of truth, similar in spirit to retrieve-and-read pipelines in open-domain QA. In case the initial summary is lacking clarity or completeness, the pipeline allows for an optional refinement step (e.g., passing the draft through another prompt or applying minor edits), though our main evaluations focus on the one-shot output. Overall, this prompt engineering strategy gives us control over the summary's content (via selected facts), structure (via IRAC labels), and length, making the generation process more transparent and tunable compared to a vanilla LLM prompt.

# 4 Experimental Setup

To validate the effectiveness of RuleSum, we designed a comprehensive evaluation on a real-world legal summarization dataset with multiple controlled variations. We used the MultiLexSum dataset, which consists of long U.S. case documents paired with expert-written summaries and annotated issue–holding pairs. From this corpus, 75 cases were sampled for evaluation (the same split as in prior work): 50 cases for development (parameter tuning) and 25 cases held out for testing. Each reference summary in MultiLexSum serves as a gold standard to evaluate content coverage and writing quality.

**Evaluation dimensions:** We systematically evaluate our system across several dimensions to isolate the impact of each component. For each test case, we generated summaries under a full factorial combination of conditions:

*Knowledge Graph Integration:* IRAC-guided KG vs. Unstructured KG. In the IRAC-guided setting, the input triples are tagged with IRAC roles (Issue/Rule/Application/Conclusion) during extraction; in the unstructured setting, no such labels are used (the triples are factual only). This tests the effect of injecting the IRAC reasoning structure into the pipeline.

*Triple Selection Policy:* IRAC-Quota vs. Similarity-based. We compare the balanced selection of facts across IRAC categories to a purely relevance-driven selection by semantic similarity. This dimension shows whether emphasizing reasoning coverage trades off any relevance in content selection.

*Serialization Format:* Free-form sentences vs. KAPING tuples vs. IRAC-labeled (KAPING-IRAC). This evaluates how the format of injected knowledge affects the summary. Free-form provides a fluent context, tuple format provides a compact factual list, and IRAC-labeled provides a structured outline.

*Target Summary Length:* Tiny ( 50 words) vs. Short ( 150 words) vs. Long ( 300 words). By varying the allowed length, we examine performance at different compression levels – from extremely concise summaries to more detailed ones.

In addition to the above, we include a baseline condition for comparison: a zero-shot GPT-4 summary with no knowledge graph input (i.e., the model only sees the case text and is prompted to summarize it). For fairness, the baseline is generated under the same length constraints as our structured runs (tiny, short, long for each case). This yields a point of reference to quantify the benefit of RuleSum's injected structure.

Combining the factors above, each test case is summarized under all $2 \times 2 \times 3 \times 3 = 36$ structured configurations, plus the baseline variants. In total, our evaluation produced on the order of a thousand generated summaries (e.g., $48$ configurations per case $\times 25$ cases $\approx 1200$ summaries) for analysis. This exhaustive setup allows us to conduct a detailed ablation study. Specifically, we not only compare the full RuleSum pipeline to the baseline, but also evaluate partial variants where we enable only one structured component at a time (e.g., only adding the KG without IRAC labels, or only using IRAC prompting without the KG, etc.). These ablation experiments help in attributing performance gains to specific components of the pipeline (KG, IRAC structure, selection policy). All generation used the same underlying model (GPT-4) and prompting approach, to control for variability.

| Metric | Variant | KG | SER | TP |
|---|---|---|---|---|
| SBERT (↑) | Ours vs. Doc | 0.714 | 0.719 | 0.715 |
| | Expert vs. Doc | 0.620 | 0.620 | 0.620 |
| | Baseline vs. Doc | 0.713 | 0.719 | 0.714 |
| FKGL (↓) | Ours | 13.04 | 12.88 | 13.07 |
| | Baseline | 13.27 | 13.27 | 13.27 |
| | Expert | 12.94 | 12.94 | 12.94 |
| | Doc | 13.93 | 13.93 | 13.93 |

Table 1: Semantic similarity and readability (SBERT, FKGL) across knowledge graph (KG), serialization (SER), and top-$k$ (TP) configurations. Higher is better for SBERT; lower is better for FKGL.

| Metric | Variant | KG | SER | TP |
|---|---|---|---|---|
| ROUGE-L (↑) | Ours | 0.192 | 0.191 | 0.192 |
| | Baseline | 0.193 | 0.193 | 0.193 |

Table 2: Lexical overlap (ROUGE-L) across KG, SER, and TP. Higher is better.

**Evaluation metrics:** We assessed each generated summary along three primary metrics that capture complementary aspects of summary quality:

*ROUGE-L (F-measure):* This metric measures lexical overlap between the generated summary and the reference summary, focusing on the longest common subsequence. ROUGE-L provides a sense of how much of the important content (as written by experts) is captured in the model's summary. A higher ROUGE-L indicates better coverage of reference facts or phrases. We report ROUGE-L as a percentage, with higher being better for content fidelity.

*SBERT-based Similarity:* To evaluate semantic fidelity, we use Sentence-BERT (SBERT) embeddings to compute the similarity between each generated summary and the source document. This embedding-based score (cosine similarity) reflects how well the summary preserves the meaning of the original text, beyond exact word overlap. We compare our summaries' SBERT scores to those of the baseline and even the expert summaries, to understand the semantic retention. Higher SBERT scores indicate that the summary contains information semantically closer to the full document.

*Flesch–Kincaid Grade Level (FKGL):* We measure the readability of the summaries using the FKGL readability test, which approximates the U.S. grade school level required to understand the text. A lower FKGL score means the summary is easier to read (simpler vocabulary and sentence structure). This metric is crucial for our goal of accessible summaries – we want legal summaries that a broader audience can comprehend. We compute FKGL for each summary and also compare against the reference and source texts as benchmarks. For instance, the original cases often have high FKGL (legal jargon), while good summaries should ideally have a lower FKGL for accessibility.

All metrics are computed for each system configuration and length. We do not rely on any single metric; rather, we consider the trade-offs – e.g., does a structured method improve ROUGE (content) and SBERT (meaning) without hurting FKGL (readability)? The evaluation is designed to answer such questions. Importantly, we refrain from hand-tuning the summaries for these metrics; all outputs are model-generated under the specified conditions, ensuring an apples-to-apples comparison. In the next section, we will present the results of these experiments, highlighting how RuleSum performs under different settings and which combination of knowledge injection strategies proves most effective.

## 5 Results

We summarize outcomes for our framework described in Figure 1, comparing the zero-shot baseline, single-component ablations, and the best combined variant. Aggregate metrics and ablations appear in Tables 1 and 2; difficulty-stratified and length-sensitivity trends are shown in Figs. 2 and 3.

**Semantic fidelity and readability across lengths.** Figure 3 shows that structured prompting preserves or improves semantic fidelity (SBERT cosine to the source document) relative to both the baseline and expert references across all target lengths, with the largest margin at *tiny* (50 words), where gains reach up to +0.12 SBERT over gold. In parallel, the same figure indicates lower FKGL (better readability) than the baseline for all lengths, suggesting that the injected structure guides

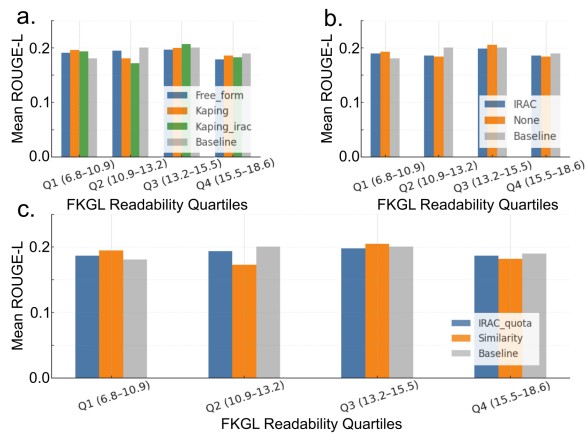

Figure 2: **Quartile analysis (by source FKGL).** Structured variants outperform the zero-shot baseline overall; KAPING_IRAC+IRAC-quota leads on harder texts (Q3–Q4).

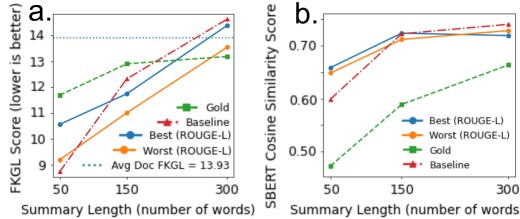

Figure 3: **Length sensitivity.** (a) FKGL (lower is better). (b) SBERT cosine similarity to source (higher is better). IRAC KG + KAPING_IRAC + IRAC-quota yields the strongest overall trade-off.

the model toward clearer phrasing rather than harming fluency. As length increases from *tiny → short → long*, SBERT rises as expected and FKGL increases slightly, yet structured variants retain a readability edge.

**Difficulty-stratified performance.** To assess robustness under varying case complexity, we stratify by FKGL quartiles of the *source* documents (Fig. 2). On easier cases (Q1–Q2), methods are broadly comparable; on harder cases (Q3–Q4), the KAPING_IRAC+IRAC-quota configuration yields the strongest lexical alignment, improving ROUGE-L by up to $+0.006$ over the baseline. This pattern indicates that explicit reasoning-role scaffolds are most helpful when language is dense, aligning with the intuition that structure mitigates complexity.

**Ablation trends and complementary effects.** Tables 1 and 2 summarize aggregate trends when toggling individual components (KG ruleset, serialization, selection policy). The full configuration—IRAC-guided KG + KAPING_IRAC serialization + IRAC-quota selection—achieves the best overall trade-off (e.g., FKGL $\downarrow 13.04$ vs. 13.27 baseline; SBERT $0.714$ vs. $0.713$ baseline). Removing any one component slightly reduces both semantic similarity and readability, indicating complementary contributions: (i) IRAC tags organize evidence by rhetorical role, (ii) IRAC-quota maintains reasoning-role coverage to prevent omissions, and (iii) KAPING_IRAC communicates that organization explicitly at inference time.

# 6   Conclusion

Our structured prompting approach significantly boosts both fidelity and clarity in legal case summaries. Across all target lengths, the IRAC-guided method outperforms a zero-shot GPT-4 baseline and even surpasses expert-written summaries in semantic similarity, achieving up to +0.12 higher SBERT cosine similarity at the shortest length. Crucially, these gains come without sacrificing readability: our outputs maintain lower Flesch–Kincaid Grade Levels than the baseline, indicating clearer, more fluent phrasing. The benefit of this structured strategy is most pronounced on com-

plex, high-density texts, where it yields stronger ROUGE-L overlap with the source and avoids the omission or misalignment errors that plague the baseline. Finally, ablation studies confirm that each component of our pipeline (IRAC-tagged knowledge graphs, structured kaping-irac serialization, and an IRAC-based quota) contributes synergistically. We show that removing any one component degrades performance, underscoring the value of incorporating domain-specific structure to guide large language models.

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
