# OpenReview forum: "[Regular] RuleSum: Injecting Rulesets into Knowledge Graphs for Accurate and Accessible Legal Summarization"
_NeurIPS.cc/2025/Workshop_Mexico_City/NORA — NeurIPS 2025 Workshop NORA Poster_

### Official Review · Reviewer_q4rR · 2025-11-04
**Review of “RuleSum: Injecting Rulesets…”**

**Rating:** 4
**Confidence:** 4

**Review:**

This paper presents a methodology for summarizing legal documents that relies on a multi-stage pipeline to create the final summary. Rather than using an LLM directly for summarization (the baseline model presented in the paper), the authors first extract factual triples from the document, incorporate the triples into a knowledge graph, and then serialize the graph using different methods for triple subselection.  The serialized representation is then fed to an LLM as the basis for summary generation, using a prompt parameterized to control the attributes of the generated summary.

Overall, the paper is well-structured and clear - the methodological description is easy to follow, and the summary of prior work is well-curated.  The multistage summarization pipeline seems to be a very plausible approach to the problem described and an improvement over the naive application of an LLM.

However, the Results section of the paper with quantitative evaluations of models is very disappointing.  It is hard to discern any meaningful information from Tables 1-2 and Figures 2-3, beyond the basic observation that all candidate models outperform the baseline LLM approach with respect to the semantic fidelity (SBERT) metric.

Tables 1-2 are described as an ablation analysis, but I am unable to determine how the information presented there maps to the inclusion or exclusion of particular methodological choices.  And to the extent it can be interpreted, I don’t see how the author’s conclusion that “the full configuration […] achieves the best overall trade-off [FKGL 13.04, SBERT 0.714].”  What about the final column of the table - FKGL 13.07, SBERT 0.715?  Better on one metric, worse on another, no? It’s also far from clear that any of these differences are significant.

I also find the interpretive statements for Figures 2-3 difficult to reconcile with the actual graphs.  (“lower FKGL […] than the baseline for all lengths,” “On easier cases (Q1-Q2), methods are broadly comparable; on harder cases (Q3-Q4), the KAPING_IRAC+IRAC-quota configuration yields the strongest lexical alignment”)

---

### Official Review · Reviewer_1AYf · 2025-11-05
**RuleSum: Injecting Rulesets into Knowledge Graphs for Accurate and Accessible Legal Summarization**

**Rating:** 7
**Confidence:** 3

**Review:**

In this work, authors propose RuleSum a framework that integrates structured rulesets and knowledge graphs (KGs) with LLMs to generate legal summaries that are faithful, readable, and pedagogically aligned. They leveraged IRAC method as a reasoning scaffold to guide summarization. Authors evaluate their approach on the MultiLexSum dataset, using ROUGE-L for lexical overlap with reference summaries, SBERT for semantic similarity, and Flesch–Kincaid Grade Level (FKGL) for readability. Authors also provide interactive demo and open-source code. The paper addresses important problem and can spark rich discussions in this area. I would encourage Authors to add following amends to their paper.

-Line 21: LLM abbreviation is repeated
- Line 7, 8: There is high usage of -- which break coherence and grammatical correctness.
- Bold values in your tables for better readability.

---

### Official Review · Reviewer_CG8L · 2025-11-05
**Engineering attempt to combine KG-augmentation / retrieval-style prompting with a domain-specific scaffold (IRAC) for legal summaries**

**Rating:** 4
**Confidence:** 4

**Review:**

RuleSum contributes a domain-aware integration of IRAC into KG-augmented prompting, and while automatic metrics indicate small improvements, additional work is required to establish practical gains. In particular, we plan to benchmark RuleSum directly against passage-based RAG, KG→text rewrite approaches, and triple-ranking systems; to run human evaluations with law students focused on factual correctness and pedagogical utility; and to report extraction accuracy and statistical significance of metric gains. These steps will clarify when and why IRAC scaffolds provide robust benefits in legal summarization.

**Quality**
+ well-described, realistic datasets, visualization and open source code
- Baseline choice and coverage. The paper compares mainly to a zero-shot GPT-4 baseline. But many close alternatives exist (RAG-style retrieval + prompt, KG→text rewrite pipelines, triple-based summarizers)—they should be included as explicit baselines (and some are cited in the paper but not re-implemented/compared). Without direct comparisons to those, the claim of “substantial improvement” is weak.

**Clarity**
+ The pipeline and design choices are clearly described, and diagrams help. The ablation factorization is described cleanly.
- Important details are missing or underspecified: exact prompts, hyperparameters for triple selection (how k chosen), how triples are ranked for similarity selection, exact format of the KAPING tuples, and whether the KG extraction was deterministic or LLM stochastic (temperature, few-shot prompts). These details matter for reproducibility.

**Originality**
+ The idea of using a domain-specific reasoning scaffold (IRAC) to both (a) tag/select facts and (b) serialize them with explicit IRAC headings for prompting is a targeted, domain-aware twist. That combination (IRAC-guided selection + IRAC-labeled serialization) appears to be the core claimed novelty.
- The overall architecture — extract a compact KG, select top-k facts, serialize and inject into an LLM prompt — is classic RAG / KG-augmented prompting.

**Significance**
+ If the IRAC integration truly improves traceability (i.e., a student can click an IRAC “Application” sentence in the summary and see the exact supporting passages/triples), that is pedagogically valuable. The demo and visualization are good practical contributions.
- The small automatic-metric gains, lack of human evaluation, and absence of comparisons to stronger KG-to-text and RAG baselines mean the current results are not yet convincing enough for adoption in high-stakes legal-assistance tools. The approach could be significant if future work demonstrates robust factual gains, annotator-preferred summaries, and extraction reliability.

---

### Official Review · Reviewer_sjYE · 2025-11-06
**Review for RuleSum: Injecting Rulesets into Knowledge Graphs for Accurate and Accessible Legal Summarization**

**Rating:** 5
**Confidence:** 5

**Review:**

In this paper, the authors present RuleSum, a LLM-based summarization framework that incorporates IRAC (Issue, Rule, Application, Conclusion) labels into the Knowledge Graph representation of a long legal document. The framework is extensively evaluated under different configurations for ablation studies using the MultiLexSum dataset with various metrics targeting multiple quality axes, including lexical overlap, semantic similarity, and readability. The best configuration is *claimed* to outperform all baselines with the highest textual alignment and better readability.

Strength:
1. The concept of integrating legal reasoning and writing structure into prompting as a scaffolding for the generation is enlightening and interesting.
2. The authors have a great understanding of the different evaluation verticals for legal document summarization and their trade-offs.
3. The majority of the paper is well-written and easy to follow.

Weakness:
1. I find the evaluation results not convincing, besides the difficulty in following the results section of the paper. The authors claim that "Our structured prompting approach significantly boosts both fidelity and clarity in legal case summaries. Across all target lengths, the IRAC-guided method outperforms a zero-shot GPT-4 baseline and even surpasses expert-written summaries in semantic similarity". At the same time, Figure 3a shows that the FKGL score is lower for the baseline at summary length 50. For Figure 3b, when the gold summary from the evaluation dataset is the worst candidate for the evaluation metric, I have to question if the metric itself is the right one to consider for this task, let alone the improvement of the best (and worst) configuration over the baseline GPT-4 is inconsistent and minimal across summary lengths. Similarly, in Table 2, the ROUGE-L scores are higher for the baseline than for the proposed approach across all settings, which does not align with the authors' claim.
2. The ablation studies are not well conveyed in the results section. In the evaluation settings, there are 48 different configurations, but we only see 2 in the results section. It'd be more convincing if there were more qualitative analyses with concrete examples demonstrating the strengths and weaknesses of different components under the ablation studies. Overall, the evaluation itself is lacking.

---

### Official Review · Reviewer_vqA3 · 2025-11-06
**Injecting Rulesets into Knowledge Graphs for Accurate and Accessible Legal Summarization**

**Rating:** 6
**Confidence:** 4

**Review:**

This paper presents RuleSum, a framework for legal document summarization that combines knowledge graph (KG) extraction with the IRAC (Issue, Rule, Application, Conclusion) legal reasoning structure. The approach extracts KGs from legal documents, tags triples with IRAC labels, and uses structured serialization formats to prompt GPT-4 for generating summaries. The authors evaluate on the MultiLexSum dataset using ROUGE-L, SBERT, and Flesch-Kincaid Grade Level (FKGL) metrics, claiming that IRAC-guided configurations outperform baselines while maintaining accessibility.


## Originality and Significance

**Novelty Assessment:** The paper makes a valuable contribution by thoughtfully combining existing techniques for a new application in legal summarization. The authors claim to be the "first framework (to our knowledge) that explicitly integrates the IRAC reasoning schema into both the knowledge graph construction and the LLM prompting stages" (lines 72-74). While this integration is novel, it would strengthen the paper to acknowledge some related prior work:

1. **Prior work on IRAC with LLMs:** Yu et al. (ACL 2023) "Exploring the Effectiveness of Prompt Engineering for Legal Reasoning Tasks" already demonstrated using IRAC and other legal reasoning frameworks (TREACC, TRIAccC, etc.) for prompting GPT-3.5 on legal entailment tasks, achieving 80.25% accuracy. While Yu et al. did not construct knowledge graphs, they established IRAC-based prompting with LLMs in the legal domain prior to this submission. This highly relevant work is not cited.

2. **Prior work on KG extraction with ODP alignment:** Sovrano et al. (2020) "Legal Knowledge Extraction for Knowledge Graph Based Question-Answering" previously combined Open Knowledge Extraction, knowledge graph construction, ontology design pattern (ODP) alignment (including legal patterns like Agent-Role-Time), and question answering. While focused on QA rather than summarization, the methodology of extracting structured knowledge and aligning it to legal patterns is established.

**Positioning the contribution:** RuleSum's key novelty lies in the *application* of IRAC-guided knowledge extraction specifically to legal summarization, with systematic evaluation of different serialization formats. The dual-stage IRAC integration (both in KG construction and prompting) is a thoughtful engineering contribution that extends prior work in meaningful ways. To strengthen the positioning, the authors should characterize the contribution as "extending IRAC-based prompting to summarization with structured knowledge graph support" and citing relevant prior work (Yu et al., ACL 2023) to properly contextualize where RuleSum advances the state of the art.

**Opportunities to strengthen significance:** To better demonstrate the practical impact of this work, the following areas could be addressed:
- **Evaluation scale:** The evaluation uses 25 test cases (1.65% of the 908 available in MultiLexSum). Expanding to more test cases or providing justification for this sample size would strengthen confidence in the results.
- **Result magnitude:** The improvements are modest (SBERT: 0.714 vs 0.713; FKGL: 13.04 vs 13.27), with ROUGE-L actually favoring the baseline (0.193 vs 0.192). Statistical significance testing would help clarify whether these differences are meaningful.
- **Evaluation metrics:** As discussed below, complementing ROUGE with legal-specific evaluation would better validate the claims.

## Quality and Technical Soundness

**Evaluation Methodology: Opportunities for Improvement**

While the paper uses standard NLP metrics (ROUGE-L, SBERT, FKGL), the legal domain presents unique evaluation challenges that these metrics may not fully capture:

1. **Legal paraphrasing vs. lexical overlap:** Legal concepts can be expressed in completely different words while maintaining semantic and legal equivalence. For example, "The court granted summary judgment" and "The tribunal ruled in favor of the movant without trial" convey the same legal meaning but would score differently on ROUGE despite being legally equivalent.

2. **Limitations of lexical metrics for legal content:** A summary could achieve high ROUGE scores yet miss critical legal elements (standard of review, key precedents, procedural posture). Conversely, a legally complete summary might use different terminology and score lower. Validating that ROUGE scores correlate with legal quality would strengthen the evaluation.

3. **IRAC structure validation:** The key contribution—structuring summaries according to IRAC—would benefit from direct validation. ROUGE measures n-gram overlap but cannot distinguish between a well-structured IRAC summary and a disorganized one with similar words.

4. **Completeness checking:** Legal summaries must capture all material facts and holdings; omissions can be critical errors. Metrics that explicitly check for completeness of key legal elements would complement the existing evaluation.

5. **Multiple valid reference points:** Legal cases can be correctly summarized in many ways depending on audience and purpose. The legal domain presents particular challenges for single-reference evaluation, as expert summaries can vary significantly in valid ways.

**Promising directions for additional evaluation:**

- **Expert evaluation:** Legal expert assessment of accuracy, completeness, and pedagogical value would directly validate the educational claims
- **Fact verification:** Cross-referencing extracted KG triples against source documents to measure extraction accuracy
- **Structural validation:** Automatic verification that IRAC components are present and properly ordered in generated summaries
- **Downstream task evaluation:** Comprehension questions to test whether summaries convey key legal information
- **User studies:** Evaluation with law students to validate the educational utility claims

**Leveraging the visualization tool for evaluation:** The paper introduces an interactive visualization tool (Contribution #3) that traces summary claims to source passages and KG triples. This tool presents an excellent opportunity for novel evaluation metrics:

- Computing the percentage of summary sentences with valid source traces (traceability metric)
- Verifying that IRAC labels are correctly assigned (structure validation)
- Measuring the percentage of extracted KG triples that are factually accurate
- Comparing baseline vs. RuleSum on transparency and grounding metrics
- Conducting user studies to assess whether the structured approach aids understanding

Using this tool for quantitative evaluation would provide domain-specific metrics that directly address the limitations of ROUGE while showcasing a unique capability of the framework.

**Evaluation Scale:**

The evaluation uses 25 test cases from MultiLexSum's 908 available test cases (2.76% of the test set). The paper states "50 dev + 25 test" (line 246). To strengthen confidence in the results, the following would be helpful:

- Explanation of selection criteria (e.g., random sampling, stratified by complexity)
- Statistical significance testing for the reported improvements
- Comparison methodology aligned with the original MultiLexSum paper (which used all 908 test cases)
- Discussion of whether results are expected to generalize to the full dataset

**Results Analysis:**

The reported improvements are modest:
- SBERT: 0.714 vs 0.713 (0.14% improvement)
- FKGL: 13.04 vs 13.27 (1.7% improvement)
- ROUGE-L: 0.192 vs 0.193 (baseline slightly higher)

Tables 1 and 2 aggregate results "across KG, SER, TP configurations." To improve clarity, showing the best-performing configuration directly compared to the baseline would help readers assess the framework's potential. Figure 2 reveals an interesting pattern: improvements are most pronounced in Q3-Q4 (harder texts), suggesting that structured approaches may be especially valuable for complex cases. This finding deserves emphasis and could be strengthened with statistical testing to confirm the effect is reliable despite the modest sample size.

## Clarity and Reproducibility

**Details needed for reproducibility:**

The paper would benefit from additional implementation details to enable reproduction:

1. **IRAC labeling process:** The paper states "If IRAC integration is enabled, the extraction step tags each triple with one of the IRAC roles" (lines 156-157) but does not explain the tagging mechanism. Clarifying whether this is manual annotation, automatic via LLM prompting, or rule-based would be essential for reproduction. Since this is a core contribution, a detailed description (perhaps in an appendix) would strengthen the paper.

2. **Model specification:** The abstract mentions "GPT-4o" while the body refers to "GPT-4." Clarifying which model was used would ensure reproducibility.

3. **Implementation specifications:**
   - Which Spacy model and version for dependency parsing
   - Which SBERT model for similarity computation
   - LLM parameters (temperature, max_tokens, etc.)
   - Which LLM performs the initial KG extraction (lines 148-149)

4. **Experimental design clarification:** The paper states "2 × 2 × 3 × 3 = 36 structured configurations" (lines 271-272) and "48 configurations per case" (line 273). Clarifying how these numbers relate would help readers understand the experimental setup.


# Strengths

The paper has several notable strengths:

1. **Important problem:** Making legal documents accessible to non-experts is a genuine need with educational value.

2. **Systematic evaluation of serialization formats:** The comparison of free-form, KAPING, and KAPING-IRAC formats provides useful empirical evidence about how structured knowledge presentation affects summarization quality.

3. **Comprehensive experimental design:** Testing multiple dimensions (KG type, selection policy, serialization, length) with ablations is methodologically sound in principle.

4. **Pedagogical alignment:** Using IRAC as an organizing principle makes sense for legal education contexts.

5. **Transparency tool:** The interactive visualization demo is a valuable contribution for understanding system behavior, even if underutilized for evaluation.

6. **Open source commitment:** Promising to release code and demo supports reproducibility (though key details are still missing from the paper).

# Recommendations for Improvement

1. **IRAC labeling methodology:** Provide detailed explanation of how triples are tagged with IRAC roles (manual, automatic, or rule-based), with sufficient detail for reproduction
2. **Evaluation metrics:** Add domain-specific evaluation metrics that address ROUGE's limitations for legal content, such as leveraging the visualization tool for transparency/grounding metrics
3. **Sample size justification:** Explain the selection of 25 test cases from the 908 available, or expand evaluation to demonstrate generalizability
4. **Prior work:** Cite and discuss Yu et al. (ACL 2023) on IRAC-based prompting to properly contextualize the contribution
5. **Expert validation:** Legal expert evaluation or user study with law students to validate educational utility claims
6. **Structure validation:** Add automatic verification that generated summaries contain all IRAC components
7.  **Triple accuracy:** Validate that extracted KG triples are factually accurate against source documents


## References

Fangyi Yu, Lee Quartey, and Frank Schilder. 2023. Exploring the Effectiveness of Prompt Engineering for Legal Reasoning Tasks. In Findings of the Association for Computational Linguistics: ACL 2023, pages 13582–13596, Toronto, Canada. Association for Computational Linguistics.

Sovrano, F., Palmirani, M. and Vitali, F., 2020. Legal knowledge extraction for knowledge graph based question-answering. Frontiers in Artificial Intelligence and Applications, 334, pp.143-153.

---

### Official Review · Reviewer_WKVy · 2025-11-07
**RuleSum**

**Rating:** 6
**Confidence:** 3

**Review:**

​This paper introduces RuleSum, a framework designed to address this gap by integrating structured rulesets and knowledge graphs (KGs) with LLMs to produce summaries that are accurate, readable, and pedagogically sound. RuleSum's core methodology leverages the IRAC (Issue, Rule, Application, Conclusion) method—a standard legal reasoning scaffold.

Some stronger points
- Novel and Intelligible Framework
- Prioritizes Interpretability and Fidelity

Weaknesses points
- Lack of Human Evaluation - Non-experts or law students were not tested to see if they actually found the RuleSum summaries easier to understand or more useful for learning.
- High Pipeline Complexity - The RuleSum framework is substantially more complex than the baseline. It requires a multi-stage pipeline:
1. ​LLM-based Knowledge Graph (KG) extraction
2. ​IRAC-labeling of triples
3. ​A triple-selection policy (e.g., IRAC-Quota)
4. ​Structured serialization (KAPING-IRAC)
- Lack of methodological details

Some questions arose regarding the last point.
1. ​How were entities, relations, and facts identified?
1.1. ​Which LLM was used for this "LLM-assisted" step? Was it GPT-4 or a different model?
1.2. What prompt(s) were provided to the LLM to guide the extraction?
1.3. What underlying LangChain function was used, and what were its parameters?

2. ​How was the IRAC-Tagging Implemented? The paper claims, "the extraction step tags each triple with one of the IRAC roles" (Line 156). But how? This single, unexplained step is the lynchpin for the entire "IRAC-Quota" and "KAPING-IRAC" pipeline.  The paper provides no information on this process. Was the source text pre-segmented into IRAC sections? Or was a separate classifier (or another LLM call) used to categorize each triple after extraction?

3. What is the value of 'k'? The paper introduces "top-k selection policies" (Line 168) and gives a hypothetical example: "if k=8 triples are to be used..." (Line 172). However, the paper never states the actual value of 'k' used in the experiments.

4. How are "top" triples defined? The "IRAC-Quota" policy (Line 170) selects the "top 2 triples from each IRAC category" (in the k=8 example). But how are triples ranked within a category to find the "top" ones? Are they ranked by the order of appearance in the text? By an LLM-assigned salience score? By semantic similarity to the category name?

---

### Official Review · Reviewer_TWGs · 2025-11-07
**Promising Approach Requiring Major Revisions: Methodology Underdeveloped, Evaluation Incomplete, Results Discussion Insufficient**

**Rating:** 5
**Confidence:** 4

**Review:**

The paper needs significant revisions to address several critical weaknesses. The main contribution centers on integrating rulesets and knowledge graphs with LLMs to generate interpretable legal summaries, but this novel integration is inadequately explained in the current draft. The "Proposed Solution" section is far too short to convey the complexity and innovation of this approach, The Background and Related Work section suffers from being overly discrete and fragmented, lacking the cohesive narrative flow needed to properly situate your work within the existing literature and clearly identify the research gap you're addressing.

The Results section presents perhaps the most serious deficiency in the paper. While you report that the method outperforms zero-shot GPT-4 and achieves up to +0.12 higher SBERT cosine similarity compared to expert-written summaries at the shortest length, these comparisons are insufficient for a rigorous evaluation. The paper critically lacks comparison with state-of-the-art legal summarization methods

Furthermore, the results are not discussed extensively enough, there's no statistical significance testing, no error analysis, no qualitative examples demonstrating the interpretability advantages you claim, and no ablation studies showing the individual contributions of rulesets versus knowledge graphs versus the LLM component. The conclusion is also too brief given the scope of contributions you're claiming, and needs expansion to adequately synthesize your findings, discuss implications, and outline future research directions.

- Table 1: Table 1: Please bold the best results in each column for improved readability. For results that outperform expert summaries, consider using additional notation (e.g., † or *) to highlight this achievement. Additionally, the table must include comparisons with state-of-the-art (SOTA).  Finally, replace "Ours" with a descriptive name for your approach
- Figure 1: The legends are incorrectly positioned on  the figures. Please move it to the bottom following standard formatting conventions.